

# Priority effects, nutrition and milk glycan-metabolic potential drive *Bifidobacterium longum* subspecies dynamics in the infant gut microbiome

Nicholas Pucci[1], Joanne Ujčič-Voortman[2], Arnoud P. Verhoeff[2,3] and Daniel R. Mende[1]

[1] Department of Medical Microbiology and Infection Prevention, Amsterdam University Medical Centers, Amsterdam, Netherlands
[2] Sarphati Amsterdam, Department of Public Health Service Amsterdam, Amsterdam, Netherlands
[3] Department of Sociology, University of Amsterdam, Amsterdam, Netherlands

## ABSTRACT

**Background**. The initial colonization of the infant gut is a complex process that defines the foundation for a healthy microbiome development. *Bifidobacterium longum* is one of the first colonizers of newborns' gut, playing a crucial role in the healthy development of both the host and its microbiome. However, *B. longum* exhibits significant genomic diversity, with subspecies (*e.g., Bifidobacterium longum* subsp. *infantis* and subsp. *longum*) displaying distinct ecological and metabolic strategies including differential capabilities to break down human milk glycans (HMGs). To promote healthy infant microbiome development, a good understanding of the factors governing infant microbiome dynamics is required.

**Methodology**. We analyzed newly sequenced gut microbiome samples of mother-infant pairs from the Amsterdam Infant Microbiome Study (AIMS) and four publicly available datasets to identify important environmental and bifidobacterial features associated with the colonization success and succession outcomes of *B. longum* subspecies. Metagenome-assembled genomes (MAGs) were generated and assessed to identify characteristics of *B. longum* subspecies in relation to early-life gut colonization. We further implemented machine learning tools to identify significant features associated with *B. longum* subspecies abundance.

**Results**. *B. longum* subsp. *longum* was the most abundant and prevalent gut *Bifidobacterium* at one month, being replaced by *B. longum* subsp. *infantis* at six months of age. By utilizing metagenome-assembled genomes (MAGs), we reveal significant differences between and within *B. longum* subspecies in their potential to break down HMGs. We further combined strain-tracking, meta-pangenomics and machine learning to understand these abundance dynamics and found an interplay of priority effects, milk-feeding type and HMG-utilization potential to govern them across the first six months of life. We find higher abundances of *B. longum* subsp. *longum* in the maternal gut microbiome, vertical transmission, breast milk and a broader range of HMG-utilizing genes to promote its abundance at one month of age. Eventually, we find *B. longum* subsp. *longum* to be replaced by *B. longum* subsp. *infantis* at six months of age due to a combination of nutritional intake, HMG-utilization potential and a diminishment of priority effects.

Corresponding author
Daniel R. Mende,
d.r.mende@amsterdamumc.nl

**Discussion**. Our results establish a strain-level ecological framework explaining early-life abundance dynamics of *B. longum* subspecies. We highlight the role of priority effects, nutrition and significant variability in HMG-utilization potential in determining the predictable colonization and succession trajectories of *B. longum* subspecies, with potential implications for promoting infant health and well-being.

## INTRODUCTION

Bifidobacteria reside in the human gastrointestinal tract from infancy through adulthood, persisting as common inhabitants throughout our entire lifespan. During the first six months of life in healthy, full-term, and breastfed infants, this genus emerges as the predominant bacterial group, playing a vital role in establishing the gut microbiome and promoting healthy infant development (*Sela et al., 2008*; *Nagpal et al., 2017*). Common practices, such as alternative milk-feeding strategies, cesarean section and antibiotics can alter the ability of bifidobacteria to colonize the infant gut, leading to lower abundances and prevalence (*Dominguez-Bello et al., 2010*; *Lewis et al., 2015*; *Hill et al., 2017*; *Turroni, Berry & Ventura, 2020*). Given that bifidobacteria have been associated with important functions such as pathogen exclusion (*Brinkworth et al., 2009*; *Fukuda et al., 2012*), maintaining gut barrier integrity (*Brinkworth et al., 2009*), promoting immune system development (*Henrick et al., 2021*), and carrying out essential metabolic activities (*Sakanaka et al., 2019b*; *Henrick et al., 2021*), a thorough understanding of the patterns and factors determining their colonization and persistence in the infant gut is needed.

The gastrointestinal tract of developing infants is an ever-changing environment, making gut colonization a complex phenomenon. The colonization process is influenced by host genotype, environmental and nutritional factors, as well as interactions within the microbiome or potentially a combination thereof (*Munyaka, Khafipour & Ghia, 2014*; *Levy, Thaiss & Elinav, 2015*; *Debray et al., 2022*). Another important factor is the order and timing of species arrival during the colonization of the gastrointestinal tract, often summarized under the term priority effects (*Fukami, 2015*). Previous studies have shown that bifidobacterial metabolism and timing of colonization significantly affect competitive interactions in *in vitro* and murine experiments as well as *in vivo* gut communities (*Sakanaka et al., 2019a*; *O'Brien et al., 2022*; *Ojima et al., 2022*; *Laursen & Roager, 2023*).

Breast milk or infant formula is the sole diet for the vast majority of infants and is simultaneously the main form of nutrition for the budding infant microbiome. Compared to infant formula, breast milk contains higher concentrations and a more diverse composition of multiple, indigestible human milk glycans (HMGs): human milk oligosaccharides (HMOs) and N-glycans (*Lönnerdal & Hernell, 2016*; *Hou et al., 2024*). A few bifidobacterial species, including the infant-type *B. breve*, *B. bifidum*, and *B. longum*, can

use these as their main energy source. These species possess the key enzymes (*i.e.,* specific glycoside hydrolases, GH, and uptake proteins) to break down the glycosidic bonds present in the structures of HMOs and N-glycans (*Ward et al., 2007*; *Marcobal et al., 2010*; *LoCascio et al., 2010*). However, bifidobacteria display substantial phenotypic variation in HMO/N-glycan utilization strategies, influencing their competitive interactions for growth substrates. This was also evident in murine models which demonstrated HMO supplementation to be a key determinant of bifidobacterial community assembly and succession outcomes (*Laursen & Roager, 2023*).

*Bifidobacterium longum* is among the most proficient HMG-utilizers and one of the most abundant species in the infant gut (*Sela et al., 2008*; *Turroni et al., 2012*). Five *B. longum* subspecies have been identified: the human-colonizing *B. longum* subsp. *longum* (BLlongum), *B. longum* subsp. *infantis* (BLinfantis) and *B. longum* subsp. *iuvenis* and the pig-colonizing *B. longum* subsp. *suis* and *B. longum* subsp. *suillum* (*Mattarelli et al., 2008*; *Yanokura et al., 2015*; *Modesto et al., 2023*). The genomes and metabolic potentials of the two important human-colonizing BLlongum and BLinfantis reflect their ecological niches (*Underwood et al., 2015*). Specifically, BLinfantis possesses a versatile metabolic system specialized for HMO/N-glycan-utilization, whereas BLlongum displays greater specialization in the degradation of plant-derived carbohydrates, a food source typical of more mature diets (*Ward et al., 2007*; *Marcobal et al., 2010*; *LoCascio et al., 2010*; *Kelly, Munoz-Munoz & Van Sinderen, 2021*). Recent studies have implied that the early-life colonization and succession dynamics of BLlongum and BLinfantis are influenced by priority effects and nutrition (*Laursen & Roager, 2023*; *Ennis et al., 2024*; *Selma-Royo et al., 2024*). The observation that BLlongum and BLinfantis are the predominant bifidobacterial species around one and four-six months of age, respectively, points towards a degree of phenotypic plasticity and possible genomic adaptation within these subspecies (*Frese et al., 2017*; *Ennis et al., 2024*). Hence, considering intra-specific genomic diversity, host life-history traits, and environmental heterogeneity is crucial for understanding the process of early gut colonization by *B. longum*.

Recent advances in machine learning (ML) and statistical methodology have led to a better understanding of biological systems *via* complex biological datasets (*Schmidt et al., 2022*; *Medina et al., 2022*). In combination with rich datasets that provide detailed metadata, such tools can uncover key factors and quantify their importance in dynamic systems like bifidobacterial populations in the infant gut.

Here we set out to determine the factors underlying the early-life gut colonization and succession dynamics of *Bifidobacterium longum* by applying modern ML tools. Samples from mothers and infants up to six months of age were collected in the prospective Amsterdam Infant Microbiome Study (AIMS) and sequenced. Using strain-tracking and metagenome-assembled genomes (MAGs), we dissected the pan-genomic and functional characteristics of *B. longum* subspecies in relation to human milk glycan-utilization (HMOs and N-glycans) and early-life gut colonization. Employing statistical modeling and ML we identified important features associated with *B. longum* abundance dynamics, accounting for priority effects, nutritional intake, HMG-utilization potential and microbial interactions. By integrating our results, we propose a conceptual framework that sheds light

on *B. longum* gut colonization and abundance dynamics during early-life development. Understanding these dynamics will provide crucial insights for promoting infant health and well-being.

## MATERIAL AND METHODS

### Amsterdam Infant Microbiome Study cohort.

Amsterdam Infant Microbiome Study (AIMS) is a multi-ethnic, prospective birth cohort study with the general objective to establish an infrastructure that allows for (explorative) research on the development of the microbiome in relation to healthy growth and development. Pregnant women living and receiving prenatal care in Amsterdam city districts are included in the cohort. Mothers are recruited mid-pregnancy and their children are followed from birth until the age of three years. Biosamples from the children are collected at 11 time points from birth until the age of three years and from their mothers at three time points (during pregnancy, birth, and when the infant is 6 m). Also, biosamples from the partner and sibling are collected during the mother's pregnancy and when the infant is six-months-old (mo). Stool, vaginal, tongue, tooth plaque and breast milk samples are collected from the mother, while stool, tooth plaque, tongue swab samples are collected from the infant, and partner/sibling. Hence the focus of AIMS is mainly on the gut and oral microbiome. Inclusion and data collection started at the end of 2019. Methods for sampling and questionnaires received approval by the Medical Ethical Examination Commission of the Amsterdam University Medical Center (METC Amsterdam UMC, Reference number: NL64399.018.17). All participants have provided written consent. Written consent for infants was provided by parents after birth.

### Study population and sample collection

A total of 72 non-invasive stool biosamples for metagenomics shotgun sequencing were collected from 24 AIMS mother-infant pairs to be sequenced. Mother and infant samples were self-collected at home during late pregnancy (34 weeks of pregnancy) and at one and six months after birth, respectively. Mother and six-month samples were obtained by swabbing faeces collected in toilet paper during defecation. Similarly, stool samples from 1 m infants were collected directly from their diapers. After collection, samples were deposited in appropriate buffer-containing test tubes and stored in household freezers at −18/20 °C. Typically within two weeks after collection, the samples were transported in a mobile freezer (−4 °C) by trained personnel to the Amsterdam Public Health Service (GGD), and stored at −80 °C until DNA extraction. Metadata associated with maternal and infant samples were collected by means of validated questionnaires. This included information about early-life nutritional intake of the infant (milk-feeding type and frequency, solid food intake) and the mother (intake of vitamins/supplements), antibiotic and probiotic intake and other medication for all participants, housing of the mother/infant (*i.e.,* number of adults and children in household, pets, smoking), health status of both mother/infant, lifestyle factors of the mother (*i.e.,* smoking, alcohol) and demographic factors (*i.e.,* age of all participants, ethnic background, education; Table S1).

## DNA extraction and sequencing

DNA was extracted using the ZY-D4306 ZymoBIOMICS-96 Magbead DNA Kit (with the use of the KingFisherTM Flex Purification System for high throughput) following BaseClear protocols. A total of 57 out of 72 samples (mother, $n = 21$; infant_1mnt, $n = 14$; infant_6mnt, $n = 22$) yielded acceptable DNA-input concentrations for sequencing. Metagenomics libraries were further sequenced on an Illumina NovaSeq platform and yielded $2 \times 150$ bp paired-end reads.

## Metagenomics reads pre-processing

Pre-processing of raw metagenomics reads was performed using BBDuk and BBMap (*Bushnell, 2014*) (Fig. S1). Raw reads were trimmed to remove adapters, poor quality bases (Q15) and further discard reads shorter than 45bp (minlength = 45). BBMap was used to discard reads mapping the reference human genome to remove human DNA from microbial reads (minID = 0.95). The remaining microbial reads were then merged into longer, single reads using the read-pairing tool BBMerge (*Bushnell, 2014*) (min read overlap = 16). The merged content of each sample subsequently underwent quality control (FastQC; Babraham Bioinformatics—FastQC A Quality Control tool for High Throughput Sequence Data) checks to determine, among others, per sequence quality score, per base sequence quality score, QC content as well as overall basic statistics. The MultiQC (*Ewels et al., 2016*) tool further pooled the quality control output of individual samples to provide summaries of quality control per datasets. Quality-controlled reads were subsequently assembled in scaffolded contigs using the SPAdes (*Prjibelski et al., 2020*) assembler in metagenomic mode.

## MAG reconstruction

We used a multi-sample binning approach as implemented in SemiBin to generate MAGs (SemiBin 1.0, *Pan et al., 2022*). In short, for each mother-infant pair, we mapped the short reads of all samples from this mother-infant pair to a set of assembled contigs (500bp) from the same mother-infant pair *via* Bowtie2 (*Langmead & Salzberg, 2012*). The multi-sample binning mode (*SemiBin multi_easy_bin*) was then used to bin samples individually, leveraging information from samples belonging to the same mother-infant pair. MAG quality was estimated using CheckM v.1.2.0 (*Parks et al., 2015*) and GUNC v1.0.5 (*Orakov et al., 2021*), and all genomes were taxonomically classified using STAG (https://github.com/zellerlab/stag). Obtained high-quality *Bifidobacterium longum* MAGs were further passed through the *anvi-estimate-genome-completeness* program to estimate the completeness and redundancy of domain-level single-copy core genes; draft genomes were considered suitable for downstream analyses when displaying ≥90% completeness and ≤5% redundancy (*i.e.,* high-quality) or ≥85% & ≤5% redundancy (*i.e,* medium-high quality).

## Public metagenomic data acquisition

We made use of four publicly available datasets to conduct *B. longum* quantitative and genomic analyses: (*Chu et al., 2017*; *Ferretti et al., 2018*; *Shao et al., 2019*; *Yassour et al., 2018*). Similar to the AIMS dataset, these studies have longitudinal sampling design

focused on healthy early-life gut microbiome colonization dynamics and development and include samples from the mothers. For all datasets, mother-infant samples were selected based on delivery mode (*i.e.,* vaginal), pregnancy length (*i.e.,* full-term), health status (*i.e.,* healthy), antibiotics use (*i.e.,* no antibiotics) and infant age (∼one-month; *Chu et al., 2017*: 40 days; *Ferretti et al., 2018*: 30d; *Shao et al., 2019*: 21d, *Yassour et al., 2018*: 30d, no matching 6 months samples could be obtained). In addition, a minimal requirement for inclusion was available metadata on milk-feeding. No public metagenomics dataset with data collected at six months of age met our criteria for inclusion. Metagenomics raw reads for the *Chu et al. (2017)*, *Ferretti et al. (2018)*, and *Yassour et al. (2018)* were downloaded from public sources and underwent the same pre-processing, assembly and *B. longum* MAG reconstruction protocol as AIMS samples. If the *B. longum* MAGs obtained from these studies were determined to be of low-quality, we retrieved the corresponding high-quality MAGs from the Early-Life Gut Genomes (ELGG) database (*Zeng et al., 2022*). For *Shao et al. (2019)*, *B. longum* MAGs were exclusively downloaded from the ELGG database. MAGs contained in this database were reconstructed using three metagenomic binning tools (MetaBAT v2.12.1, *Kang et al., 2019*), MaxBin v2.2.6, (*Wu, Simmons & Singer, 2016*) and CONCOCT v1.0.0 (*Alneberg et al., 2014*) and quality-controlled with CheckM v1.0.12 and GUNC v1.0.5. Only MAGs with ≥90% completeness and ≤5% redundancy were downloaded from the ELGG database. For all datasets, additional information on vertical transmission events and nutritional intake was obtained *via* metadata made available by *Podlesny et al. (2022)*. Across datasets, the percentage of MAGs failing quality control was 27% for AIMS samples and on average 69% for the publicly available studies. In total, 125 metagenomics mother and infant stool samples and 34 *B. longum* MAGs were obtained after quality-filtering. Processed metagenomics reads and MAGs were used for quantitative profiling of *Bifidobacterium* spp. as well as meta-pangenome and CAZy-profile analyses, respectively.

## Bifidobacterial (sub)species-profiling

In order to assess the relative abundance of bacterial taxonomic groups per sample and the alpha and beta diversity of the sequenced microbial communities, concatenated reads were mapped onto reference genomes from the GTDB R220 database (*Parks et al., 2020*) by using the computational tool sylph v.0.6.1 (*Shaw & Yu, 2024*). To validate the sylph abundance estimates for the *B. longum* subspecies we profiled the AIMS samples using MetaPhlAn v4.0.6 with the amended MetaPhlAn-B.infantis database (Fig. S5) (*Ennis et al., 2024*; *Blanco-Míguez et al., 2023*).

## Bifidobacterial strain-tracking

Single nucleotide variant (SNV) calling was used to determine overall and bifidobacterial strain-tracking between mothers and 1 m infants (*i.e.,* vertical transmission), mothers and 6 m infants (*i.e.,* persisted vertical transmission or horizontal transmission) and within same infants (*i.e.,* persistence). Reference-based strain tracking was performed by running StrainPhlAn4 (*Blanco-Míguez et al., 2023*). StrainPhlAn4 employs MetaPhlAn 4 (*Blanco-Míguez et al., 2023*) to map quality-controlled merged reads to species-specific

marker sequences. StrainPhlAn4 multiple sequence alignment for *Bifidobacterium* spp. of interest was performed using the *strainphlan* module on reconstructed species-specific consensus and reference markers.

### *B. longum* meta-pangenome and phylogenomics tree

Pangenome analyses were carried out in Anvi'o v8 (*Eren et al., 2021*). MAGs and reference genomes were converted into contig databases (*anvi-gen-contig-database*) and functionally annotated from COG and KEGG databases (*anvi-hmm, anvi-run-ncbi-cogs* and *anvi-run-kegg-kofams*). A genome storage was created from functionally annotated contig databases (*anvi-gen-genomes-storage*); this storage contained essential information about the genomes, which was then used as input for pangenome reconstruction. We made use of the *anvi-pangenome* and *anvi-interactive* modules to first reconstruct the *B. longum* pangenome and subsequently visualize the hierarchical clustering of MAGs based on gene copy-number (Method: Ward; Distance: Euclidean; Fig. S2).

We further used the information retrieved from our *B. longum* pangenome to reconstruct a phylogenomics tree in anvi'o (https://merenlab.org/2017/06/07/phylogenomics/). Briefly, nucleotide sequences for 172 single-copy core genes found to be present among all *B. longum* MAGs were obtained and aligned using the *anvi-get-sequences-for-gene-clusters* module. The phylogenomics tree was obtained by using the resulting concatenated alignment of nucleotide sequences as input for the *anvi-gen-phylogenomics-tree* program (Fig. S3). The genome of *B. adolescentis* ATCC 15703 was used as an outer group.

### *B. longum* MAGs subspecies differentiation

Subspecies differentiation was achieved by reconstructing a phylogenomics tree (Fig. S3) of reconstructed MAGs and *B. longum* subsp. *longum* (BLlongum) and *B. longum* subsp. *infantis* (BLinfantis) reference genomes. MAGs clustering with a specific reference genome (downloaded from the proGenomes3 database (*Fullam et al., 2023*) were subsequently assigned to the respective subspecies. We further validated the subspecies-differentiation of reconstructed *B. longum* MAGs by using the MetaPhlAn-B.infantis database (*Ennis et al., 2024*), an amended MetaPhlAn database which includes *B. longum* subspecies-specific marker genes. Briefly, gene (nucleotide) sequences derived from *B. longum* MAGs were aligned to 247 subspecies-specific marker genes (119 BLinfantis and 128 BLlongum) nucleotide sequences for the subspecies-specific marker genes (https://github.com/yassourlab/MetaPhlAn-B.infantis) using BLASTn (v2.6.0+) (*Altschul et al., 1990*) filtered at ≥90% alignment identity and ≥50% coverage (*Ennis et al., 2024*). Hierarchical clustering of *B. longum* MAGs based on marker genes copy number (distance metric = euclidean, clustering method = Ward.D2) was used to differentiate BLinfantis from BLlongum MAGs (Fig. S4, Table S2). The complete list of *B. longum* subsp. *longum* (BLlongum) and *B. longum* subsp. *infantis* (BLinfantis) MAGs used in this study are provided in Table S3.

### Functional and metabolic enrichment analyses

We used reconstructed high- and medium-quality MAGs to investigate potential functional differences between and within BLlongum and BLinfantis (Table S9–S11). Reference

genomes were excluded from functional comparisons, resulting in enrichment scores and adjusted *p*-values (*q*-value) of functions found in BLlongum and BLinfantis MAGs were obtained by using the *anvi-compute-functional-enrichment-across-genomes* module. Shortly, this method aggregates gene calls by their respective KEGG functional annotation and fits a generalized linear model (GLM) to determine the uniqueness of the association to each of the compared groups (anvi-compute-functional-enrichment-across-genomes).

Described functional annotations were used to further predict metabolic modules and pathways (KEGG MODULE resource) for BLlongum and BLinfantis. For this purpose, we used the *anvi-estimate-metabolism* module, which predicts both presence and completeness of the detected metabolic modules (*Watson et al., 2023*). A completeness score of ≥75% was applied to determine the presence of modules in any given MAG. The *anvi-compute-functional-enrichment* module was then applied to select present modules and fit a GLM with a logit linkage function to determine their enrichment in BLlongum and BLinfantis subspecies (Table S10). The same module was then applied on a set of BLlongum haplotypes in order to describe BLlongum-specific metabolic modules. As previously described, metabolic enrichments in vertically-transmitted and non-transmitted haplotypes could not be tested with the available dataset. We therefore limited ourselves to investigate and compare the presence of specific metabolic modules associated with high metabolic independence (HMI) and found to be enriched in good colonizers of fecal microbiome transplantation (FMT; Table S11) (*Watson et al., 2023*).

## Profiling of HMG-utilization potential

Predicted proteomes of reconstructed *B. longum* MAGs were annotated using the CAZy database using run_dbCAN4 (*Zheng et al., 2023*). Based on the literature, predicted proteins sequences encoding for 13 HMO- ($n = 8$) and N-glycan-utilizing ($n = 5$) glycoside hydrolases (GH) were further annotated using eggnog-mapper to identify potential variants of GH-encoding genes whenever they were not annotated to the same orthologous group at all levels (*Arzamasov & Osterman, 2022*). To further investigate these potential gene variants, we used mmseq2 (v15.6f452) (*Steinegger & Söding, 2017*) to perform pairwise protein sequence alignments for all available sequences from our MAGs for these GHs. We then used hierarchical clustering (distance metric = percent dissimilarity, clustering method = Ward.D2) to identify clusters representing GH variants. Sequence clusters were considered to represent the same GH variant if their average alignment identity was ≥60%. Singleton sequences and GHs variants with less than 5% prevalence across MAGs were not included in the characterization of HMG-utilization potential. Hence, final annotations of GH-encoding genes reflect the integration of CAZy with eggnog annotations and protein homology searches (Table S5 and Figs. S7–S14). Alongside HMG-utilizing GHs, 37 genes, collectively encoding for 11 HMO- and three N-glycan uptake proteins, were also included in the characterization of glycan-utilization potential of *B. longum* strains (Table S6). Predicted protein sequences of genes derived from *B. longum* MAGs were aligned to glycan-transporters using BLASTp (v2.6.0+). BLASTp results were filtered using bitscore (≥50), e-value (≤0.001) and percent identity (≥60%) and only the alignment with the highest percent identity ("best hit") was retained for further analysis. Statistical analyses on

predicted HMG-utilization gene profiles were performed in R. We performed hierarchical clustering (distance metric = euclidean, clustering method = Ward.D2) on scaled absolute counts of HMG-utilizing genes to investigate similarities/dissimilarities between *B. longum* MAG based on their cluster assignments (*i.e.*, HMG-profiles).

Significant differences in the abundance of GH-and transporter-encoding genes across clusters were tested by Kruskal-Wallis tests performed on log-transformed gene abundances . Pairwise Wilcoxon Signed-Rank tests (*pairwise.wilcox.test*) were used as non-parametric post-hoc tests to determine specific group differences (Table S8). Genes were considered to be specifically enriched/depleted in a profile if displaying significant differences among groups and a log2 fold change of ±2 (4-fold change in abundance) in the 'Cluster of Interest *vs.* Remaining Clusters' comparison.

## Modeling and selection of abundance-associated features

We investigated the relationship between a set of factors and *B. longum* and *Bifidobacterium* spp. abundance during early-life gut development. We performed linear nested Least Absolute Shrinkage and Selector Operator (LASSO) regressions to identify and describe the nature of important associations of selected features representative of maternal and temporal effects (*i.e.,* maternal and one-month (sub-)species abundance, vertical transmission and strain sharing events), early-life nutrition (*i.e.,* milk-feeding type and frequency), HMG-metabolic potential (*i.e.,* HMO/N-glycan-utilizing GH- and transporter-encoding gene cluster assignments; see Methods: Profiling of HMG-utilization potential) and bifidobacteria interactions (*i.e.,* bifidobacteria diversity, presence and abundance of individual *Bifidobacterium* spp.). A *Bifidobacterium* species was considered to be present or co-occurring in a sample when its relative abundance was >0.01% (*Wood & Salzberg, 2014*).

We built subspecies-specific models for BLlongum and BLinfantis subspecies. For each taxonomic group of interest, we tested our models for each age group (*i.e.,* one-month and six-months olds), and we used corresponding relative abundances as response variables. For the one-month age group, we tested the BLlongum model only as the prevalence of BLinfantis across datasets was extremely low (5%, five out of 93 samples). Feature selection varied per model and age group (Table S13) and aimed at including as many features as possible. First, we filtered the metadata as a considerable portion did not exhibit sufficient variability (*e.g.*, solid food intake, Table S1). For each subspecies and age-group, factor variables were tested for collinearities using the *cor* function from the stats package in RStudio, using a threshold of ±0.70 (Table S12). Different combinations of covariates were tested in order to select the best/reliable model; the output of the collinearity tests, a priori knowledge, (non-)uniformity, quality of the available data and best cross-validated mean squared error ('cvm') pseudo-objectively determined the choice for best model.

LASSO regressions were carried out using the *cv.glmnet* function in RStudio (*glmnet* v4.1-8 (*Friedman, Hastie & Tibshirani, 2010*). Categorical predictor variables were dummy coded and included in the input factor matrix as continuous predictor variables. In order to account for missing information, missing values were imputed by setting the *na.inpute* option to *TRUE* in the *makeX* R function. Coefficients (β) were selected at the minimum

cross-validation error (lambda min). Features for which β was not shrunk to 0 were considered as 'important', and associated with changes in the abundance of model genus, species and subspecies. Cross-validation for one month was performed using ten times nfolds = 10 and mean squared error (mse) for accuracy. Due to the small sample size, a ten times Leave One Out Cross Validation (LOOCV, nfolds = sample size $n$) was performed on six-months LASSO models, using the mean squared error (mse) for accuracy, too. It is important to note that the output of LASSO models was not used for prediction of bifidobacterial abundance, but rather for feature selection.

## Quantification of priority effects

To test if factors are indicative of priority effects and quantify them, we made use of the formula to calculate the strength of priority effects ($P'_{ij}$) developed by *Vannette & Fukami (2014)* and refined by *Debray et al. (2023)*:

$$P'_{ij} = ln \left[ \underline{B}(i)_{ji} / \underline{B}(i)_{ij} \right]$$

where $\underline{B}(i)_{ij}$ and $\underline{B}(i)_{ji}$ represent the average abundance reached by the (sub)species in question when introduced first and after other species, respectively. As the exact arrival order cannot be deduced from our samples, we define proxies for arrival order/early arrival using three different criteria: (i) the (sub)species in question is the most abundant *Bifidobacterium* in the mother (or the closest previous time point, above a threshold of 0.1%), (ii) a strain of (sub)species in question is shared between mother and infant and (iii) either (i) or (ii) or both are true. We applied the formula using all three criteria, separately. The interpretation of $P'_{ij}$ is similar among formulas; negative coefficients indicate that proxies for early arrival of BLlongum are associated with higher BLlongum abundance.

## Data analysis and statistics

Data analyses and statistical procedures were performed using a combination of the RStudio (v2022.02.0+443) and Anvi'o (v8; *Eren et al., 2021*). Pangenomic hierarchical clusterings of MAGs and their visualization, as well as functional and metabolic enrichment analyses using anvi-modules were performed in the Anvi'o 8 platform. All other statistical tests were performed using RStudio. In addition, in all instances where multiple testing occurred, false discovery rate ('fdr') correction was implemented to account for false positives.

## Data availability

The raw-reads for each AIMS biosample were submitted to the European Nucleotide Archive (ENA) at EMBL-EBI repository and the National Center for Biotechnology and Information (NCBI, https://www.ncbi.nlm.nih.gov/bioproject/1050518), with BioProject accession number PRJEB66728.

## RESULTS

### *B. longum* subspecies replacement between one and six months of age

We sequenced 57 stool samples from 24 mothers and their full-term, vaginally-born, antibiotic free infants enrolled in the AIMS study (Table S15, also see Methods) using

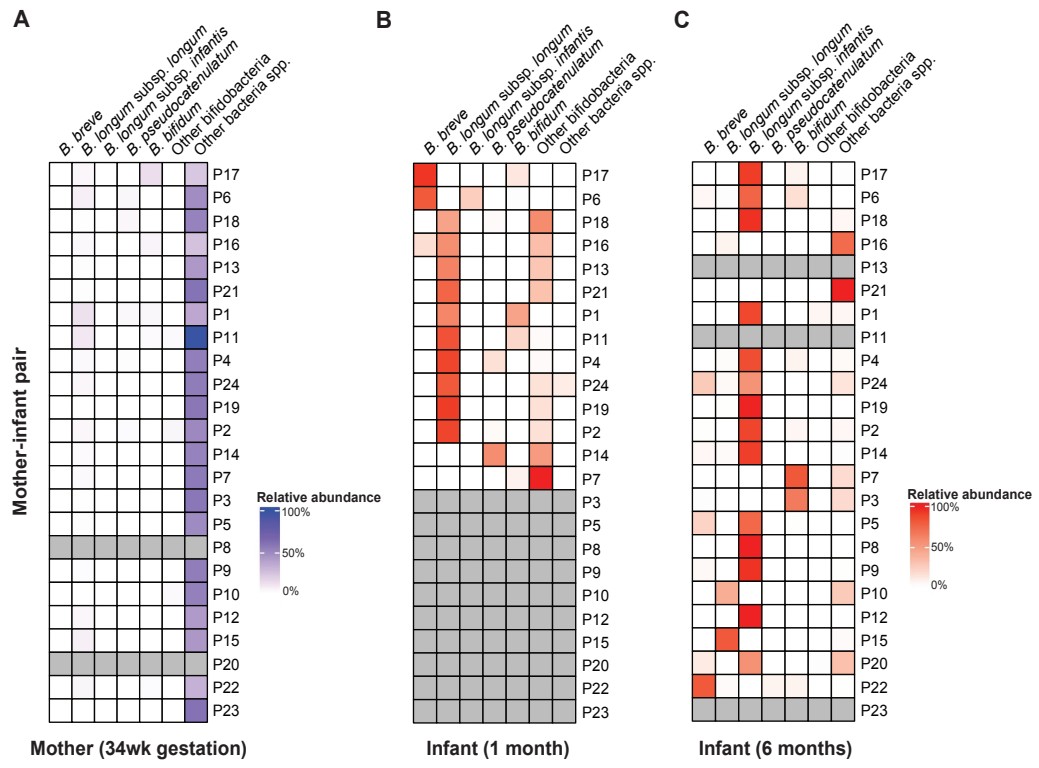

**Figure 1** ***B. longum*** **subspecies replacement between one and six months of age.** Relative abundance profiles of abundant (>10% abundance in at least one infant) bifidobacteria displayed for (A) mothers, (B) 1 m and (C) 6 m infants (mother-infant pair =P#). The columns 'Other bifidobacteria' and 'Other bacteria spp.' include non-abundant *Bifidobacterium* spp. and non-bifidobacteria taxonomic groups, respectively.

shotgun metagenomics (mean depth: 38.8M reads per sample; Fig. S1). In addition, we used 125 publicly available infant metagenomics samples.

Consistent with results from other cohorts, taxonomic composition analyses using sylph showed the *Bifidobacterium* genus to be the most abundant and prevalent taxonomic group in the gut of AIMS infants at both one and six months of age (*Frese et al., 2017*; *Shao et al., 2019*; *Ennis et al., 2024*; *Shaw & Yu, 2024*; *Kujawska et al., 2020*). Bifidobacteria accounted for ∼64% (standard deviation (SD): ±9.32%) and ∼77% (±7.25%) of the total gut microbiome, respectively (Fig. S6). Among the 13 bifidobacterial species detected, *Bifidobacterium longum* was the most abundant and prevalent species at both one and six months of age (Fig. 1, Table S15).

Across AIMS infants, BLlongum was the most abundant and prevalent *Bifidobacterium* (sub-)species at one month of age, making up 62.4% (±7.5%) of the total gut microbiome when detected (prevalence =85%). Notably, at six months of age, BLinfantis replaced BLlongum as the most abundant (79.6% ±6.7%) and prevalent (71%) taxon, even though BLinfantis was detected in only two infants at one month of age. At six months of age, BLlongum prevalence increased to 67%, though its abundance decreased when present (10.5% ±6.2%). Across the publicly available datasets, we detected a similar pattern,

which we also confirmed using MetaPhlAn (Fig. S5) (*Blanco-Míguez et al., 2023*; *Ennis et al., 2024*). BLlongum was detected to be the most abundant *B. longum* subspecies (see Methods) in maternal ($n = 48$) samples and the majority of 1 m (93%, $n = 57$) fecal samples, when present. In contrast, BLinfantis was found to be the most abundant *B. longum* subspecies in ∼7% of 1 m infants. Alongside BLlongum and BLinfantis, other *Bifidobacterium* species were also found to be present in the gut of 1 and 6 m AIMS infants (Fig. 1 and S6). Despite their lower prevalence, *B. breve* (abundance: 44.4 ±21.4%, prevalence: 31%) and *B. bifidum* (abundance: 16.6% ±7.2%, prevalence: 38%) were the second and third most abundant species in the gut of 1 m AIMS infants, when present. While displaying different abundance dynamics over time, at six months of age both species were found to be highly prevalent across the AIMS cohort (prevalence: *B. breve* =71.4%; *B. bifidum* =52%).

Mother-infant and infant-infant strain tracking revealed *Bifidobacterium* species-dependent rates of vertical transmission events and strain persistence (Table S4). We detected the presence of maternally transmitted BLlongum in ∼29% of 1 m AIMS infants (Table S4, also see Methods). Conversely, other bifidobacterial species, namely *B. pseudocatenulatum*, *B. bifidum*, and *B. adolescentis*, exhibited a higher proportion of maternally shared strains, accounting for 80%, 60%, and 80% of their occurrences, respectively. At one month of age, we did not detect shared BLinfantis and *B. breve* strains between AIMS mothers and their infants. This is congruent with our observation of BLlongum replacement by BLinfantis between the ages of one and six months and low abundance and prevalence of BLinfantis in maternal fecal samples. On the other side, 20% (three out of 15) of BLlongum occurrences in 6 m infants were the result of strain sharing with mothers. Overall, our results highlight BLlongum as the most prevalent and abundant *B. longum* subspecies in the gastrointestinal tract of 1 m infants and reveal its replacement by BLinfantis at six months of age.

## Targeted glycoside hydrolase and transporter profile analysis reveals inter- and intra-subspecies variability in HMG-utilization potential

The success of *B. longum* subspecies in the developing human gut microbiome is largely owed to their glycoside hydrolases (GH), which allows them to breakdown human-milk oligosaccharides (HMOs) and N-glycans (*Arzamasov & Osterman, 2022*). To investigate the metabolic potential of the most abundant bifidobacteria in our cohort, we used metagenomic binning to recover 63 *B. longum* MAGs (AIMS: $n = 28$, public datasets: $n =35$) and created a *B. longum* pangenome (Fig. S2). Phylogenomics analysis of *B. longum* MAGs allowed us to differentiate between BLlongum and BLinfantis MAGs (Fig. S3), which we then validated using the MetaPhlAn-B. infantis database (see Methods).

We then applied a targeted approach to profile glycoside hydrolase (GH)- and transporter-encoding genes in *B. longum* MAGs that are involved in HMG (*i.e.,* HMO and N-glycan) utilization. We define MAG HMG-utilization profiles as the presence and copy-number of these genes (Table S7). Hierarchical clustering assignments of MAGs, based on HMG-utilization profiles, were used to describe differences in HMG-utilization capacity between and within subspecies and the observed BLlongum/BLinfantis replacement. We

identified three major clusters, one of which represented BLinfantis (BLinfantis-HMG; Fig. 2A), while BLlongum was subdivided into two different HMG-utilization clusters (BLlongum-HMGsimple, BLlongum-HMGcomplex; Fig. 2A). Multiple gene enrichment analyses were then performed to capture differences (and similarities) among identified gene profiles (Fig. 2B, Table S8).

The differentiation of BLlongum MAGs into BLlongum-HMGsimple and -HMGcomplex utilization profiles was largely driven by differences in oligomannose N-glycan utilization. MAGs comprising the BLlongum-HMGsimple cluster were not enriched with a specific set of CAZymes, but harbored GH-encoding genes shared with other cluster types (*i.e.,* GH2_v1-2, v7, GH42_4, GH112 and GH136: Figs. 2B–2C). In contrast, the BLlongum-HMGcomplex cluster was enriched in four GH-encoding genes. Among these, genes encoding for alpha-mannosidases (GH38_v1, GH125) and a β-mannosidase (GH5_18) are also enriched in the BLinfantis-HMG cluster, while GH85_v1 (endo- β-N-acetylglucosaminidase) is specific to the BLlongum-HMGcomplex cluster. In BLlongum, these enzymes are involved in the intracellular degradation of oligomannose N-glycans (*Arzamasov & Osterman, 2022*). BLlongum haplotypes possessing genes encoding for GHs involved in N-glycan utilization (BLlongum-HMGcomplex) also harbored genes for N-glycan uptake (*mnaABC*), which are not found in other BLlongum haplotypes. Genes involved in the uptake of LNB/GNB (*gltABC*) were widespread across BLlongum MAGs, regardless of HMG-utilization cluster identity. Alongside the set of significantly enriched genes shared with BLlongum-HMGcomplex MAGs, BLinfantis further encoded for 14 subspecies-specific HMO- and N-glycan-utilizing GH-encoding genes and ten glycan-uptake proteins (Figs. 2A–2B). Accordingly, BLinfantis has a wide and diverse array of GH-encoding genes allowing for the cleavage of a larger diversity of glycosidic linkages, facilitating the utilization of a greater amount and diversity of HMOs and N-glycans (*Garrido et al., 2011*; *Sakanaka et al., 2019b*) (Fig. 2D). For example, the presence of alpha-L-fucosidase (GH95) is indicative of fucosylated HMO utilization (*Arzamasov & Osterman, 2022*) (Fig. 2A). Alongside their role in HMG degradation, other enriched GHs are also involved in the metabolism of N-glycans. Together with the previously described GH5_18, GH38_v1 and GH125 enzyme families, the alpha-L-fucosidase (GH29_v1-2, v4, GH95), ß-hexosaminidase (GH20_v1 and v3-4) and sialidases (GH33_v1-2) potentially play an important role in the intracellular degradation of complex N-glycans (Fig. 2D). The diversity in the GH-repertoire of BLinfantis is complemented by an increased number and diversity of HMG transporter-encoding genes. Enriched HMO transporters are potentially involved in the uptake of LNB/GNB (*gltFGH*), fucosylated HMOs (FL1-2) (*Sakanaka et al., 2019b*), LNnT and other type II HMO moieties (*nahS, hmoABC, hmoA2B2C2, hmoA3-6*) (*Kujawska et al., 2020*). BLinfantis haplotypes were also enriched in genes encoding for proteins involved in N-glycan internalization (*mnaABC* and *afcT*) (*Arzamasov & Osterman, 2022*; *Cordeiro et al., 2023*). Notably, only three BLinfantis haplotypes harbored genes encoding for the LNT/LNB/GNB transporter GltABC (Blon_2177).

We further used the *B. longum* meta-pangenome to compare genomic content between BLlongum gene profile groups and by doing so, analyze associations between HMG-utilization and overall functional potential. Our functional enrichment analysis supported

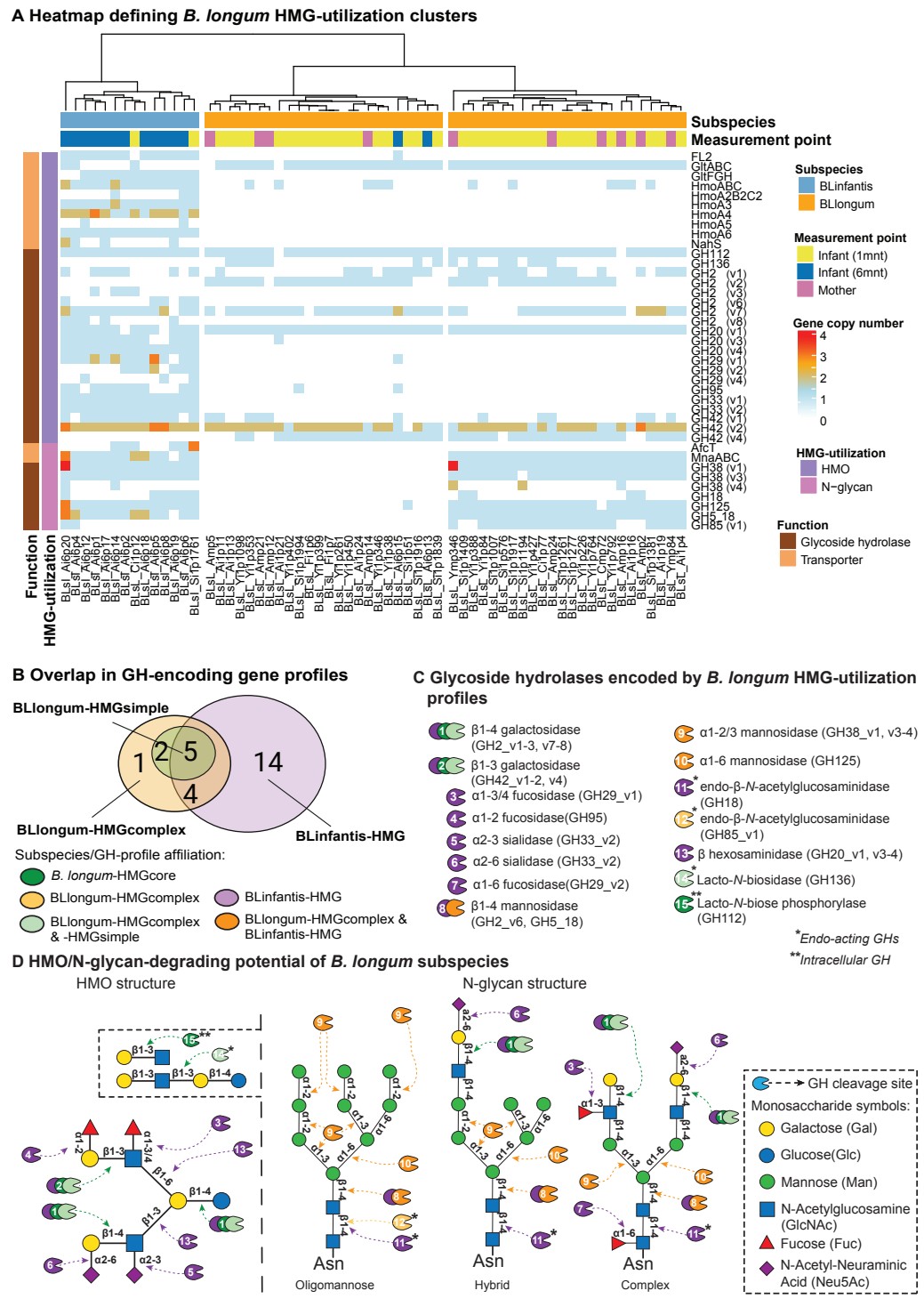

**Figure 2** **Targeted HMG-profile analysis reveals inter- and intra-subspecies variability in HMG-utilization potential.** (A) Heatmap of GH- (brown rows) and transporter (light orange rows)-encoding gene copy numbers per *B. longum* MAG (BLinfantis = light blue, BLlongum = orange; sample origin: mother = pink, 1-month infant = yellow, 6-month infant = blue). (continued on next page...)

**Figure 2 (…continued)**
For ABC transporters, only the first gene in each operon is shown, where applicable. (B) Venn diagram of GH-encoding genes among *B. longum* subspecies' HMG-profiles (C) HMO/N-glycan degrading GHs (CAZy annotations). Coloring indicates affiliation to subspecies/HMGprofiles (see B). (D) Potential hydrolytic activity of GH encoded by BLlongum and BLinfantis haplotypes. The HMO and N-glycan structures show possible linkages of HMO and oligomannose, hybrid and complex HMG building blocks, respectively (Symbols Nomenclature for Glycans, SNFG). Most enzymes depicted in this figure are exo-acting and release only terminal monosaccharide residues from non-reducing ends; in C and D, endo-acting and intracellular GH-cleaving enzymes are marked with one (*) and two (**) asterisks, respectively.

our findings from our targeted GH and transporter-profile analysis, as alpha-mannosidase (EC:3.2.1.24: GH38) and mannosyl-glycoprotein endo-beta-N-acetylglucosaminidase (EC:3.2.1.96: GH18 and GH85) are found to be significantly associated to the BLlongum-HMGcomplex gene-profile (Table S9).

Additionally, we tested for overall functional and metabolic differences between BLlongum and BLinfantis (Tables S10 and S11) and found the specific genome content of BLinfantis to reflect known differences in riboflavin, arabinose, sialic acid and urea metabolism between *B. longum* subspecies (Table S10) (*Sela et al., 2008*). Investigating associations between BLlongum genome content with strain sharing yielded no significant outcomes.

## Maternal effects, nutrition and HMG-utilization potential drive *B. longum* subspecies abundance dynamics at one and six months of age

To identify important personal, bifidobacterial, maternal and environmental factors associated with the abundance and dynamics of *B. longum* subspecies in the developing infant gut within one modeling framework, we made use of nested cross-validated LASSO regressions. At one month of age, the model was fitted on a combined dataset of infants ($n = 81$) from the AIMS ($n = 14$) and publicly available cohorts ($n = 67$), whilst six-months models were performed on AIMS infants exclusively ($n = 22$). The number of features used varied per model and age group (model: 1mnt-BLlongum $=9$; 6mnt-BLlongum $=9$; 6mnt-BLinfantis $=8$; Table S13) and aimed at maximizing the number and diversity of parameters used, while accounting for, among others, collinearity, (non-)uniformity and quality of the available data.

In order to identify factors associated with BLlongum and BLinfantis abundance across the first six months of life, we tested three LASSO models. We use two models to select features associated with BLlongum abundance at one ($n = 81$) and six months ($n = 22$). As BLinfantis prevalence was too low among 1 month infants (6mnt BLinfantis $=$five out of 93), only one model at six months ($n = 22$) was tested for this subspecies.

At one month of age, we find distinct sets of features associated with the abundance of *B. longum* subspecies (Fig. 3 & Table S14). The selected features included factors indicative of maternal effects (*i.e.,* maternal BLlongum abundance and vertical transmission), and interspecies interactions (*i.e.,* presence/absence and abundance of other bifidobacteria), as well as nutritional and genomic information. BLlongum abundance was generally positively associated with maternal effects (Fig. 3A). Interspecies interactions were found to be positive for *B. adolescentis* and *B. bifidum* and negative for total bifidobacterial species diversity,
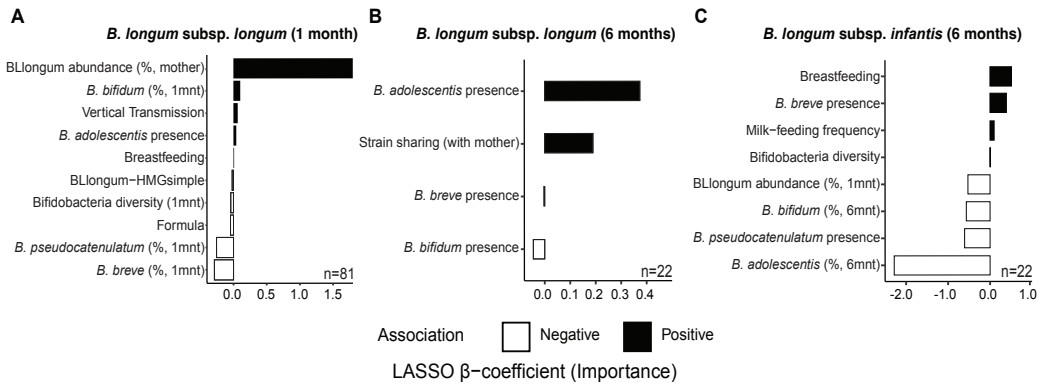

**Figure 3** **Maternal effects, nutrition and HMG-utilization potential drive *B. longum* subspecies abundance dynamics at one and six months of age.** Bargraphs show the LASSO regression cross-validated shrunken β coefficients related to important features positively (black bars) or negatively (white bars) associated to the abundance of (A) BLlongum at one month of age and (B) BLlongum and (C) BLinfantis at six months of age.

*B. breve* and *B. pseudocatenulatum* abundance. Breastfeeding and formula-feeding were positively and negatively associated with the abundance of BLlongum, respectively. Notably, HMG-utilization potential was also among the selected features; BLlongum-HMGsimple MAGs were found to be less abundant relative to BLlongum-HMGcomplex haplotypes (Figs. 3A).

BLlongum abundance at six months of age was found to be mostly associated with interspecies interactions and maternal-offspring strain sharing. Co-occurrence with *B. adolescentis* was associated with higher BLlongum abundance (Fig. 3B), while the presence of *B. bifidum* and *B. breve* was negatively associated with BLlongum abundance. Notably, haplotypes shared between mother and 6 m infants were also found to be more abundant than non-shared BLlongum.

At six months of age, BLinfantis replaced BLlongum as the most prevalent and abundant gut bacteria. Higher abundances of BLinfantis were positively associated with total bifidobacterial diversity (defined as the total number of bifidobacterial species detected in a sample) and its co-occurrence with *B. breve* (Fig. 3C). Moreover, our LASSO regression indicates breastfeeding and daily milk-feeding frequency to be important features positively associated with BLinfantis abundance at six months of age. Conversely, co-occurrence with *B. pseudocatenulatum* and increasing abundances of *B. bifidum* and *B. adolescentis* were found to be indicative of lower BLinfantis abundance at this age. Increased abundances of BLlongum at one month were found to be negatively associated with BLinfantis abundances at six months of age.

## Identification of potential priority effects

The factors we found to influence BLlongum abundance throughout the first six months of life could generally be summarized as maternal, nutritional, metabolic and competitive. Maternal factors can be indicative of early colonization, which in turn can be associated with priority effects (*Sprockett, Fukami & Relman, 2018*; *Lou et al., 2021*). To test if the
identified maternal factors acted as proxies for priority effects, we used an established framework to calculate the strength of priority effects (Methods, *Vannette & Fukami, 2014*; *Debray et al., 2023*). Maternal strain sharing and the identification of BLlongum as the most abundant *Bifidobacterium* (sub-)species were both factors that contributed to priority effects (Fig. S15), with the combination showing much stronger effects, indicating an additive relationship (Fig. S15B). These two factors were also found to be significantly associated with a higher BLlongum at one (mother/infant strain sharing: $p = 0.02$, BLlongum most abundant: $p = 0.04$, combination: $p = 0.005$) and six months of age (Mat. strain sharing: $p = 0.02$, combination: $p = 0.02$; Fig. S15A, Tables S17–18). We further compared maternal BLlongum abundance of mothers that shared BLlongum strains with their 1-month-old infants and those for which we did not detect strain sharing, but found no significant difference (Wilcoxon signed-rank test: $p = 0.08$; Fig. S16, Table S16). The estimated strength of priority effects was markedly higher at 6 months relative to one month of age, though the prevalence of BLlongum was lower at that age (Fig. 1).

In summary, by analyzing AIMS and publicly available datasets we identify a common infant gut colonization and succession trajectory for *B. longum* subspecies (BLlongum-to-BLinfantis replacement) and less common trajectories (Fig. 4), driven by maternal influence/priority effects, nutrition and HMG-utilization potential.

## DISCUSSION

In this study, we investigated the factors governing colonization and succession patterns of *B. longum* and its subspecies by combining precision metagenomics and pangenomics analyses with detailed data describing nutrition as well as maternal and environmental factors. Our analyses identified an interplay of maternal influence/priority effects, feeding patterns, and bifidobacterial metabolic potential to regulate early life *B. longum* subspecies abundance dynamics.

The role of strain transmissions from mother to child are well-documented (*Zhu et al., 2015*; *Ferretti et al., 2018*; *Valles-Colomer et al., 2023*), yet we found *B. longum* subspecies abundance in the maternal microbiome to be strongly associated with higher abundance of this subspecies in the infant. Maternal factors identified in this study as proxies for priority effects were significantly associated with BLlongum abundance at one and six months of age (Fig. S15A, Table S18). Notably, BLlongum maternal abundance and detected vertical transmissions did not significantly influence each other (Fig. S16, Table S16). This may suggest the occurrence of additional vertical transmission events that remained undetected due to biological effects related to population bottlenecks (*e.g.*, strain differentiation after transmission) (*Chen & Garud, 2022*) or technical issues related to strain tracking. Nevertheless, higher maternal BLlongum abundances may facilitate infant gut colonization. Both maternal factors were found to be indicative of priority effects and we found them to have additive effects in influencing BLlongum abundance in the infant (Fig. S15). We could not perform similar analyses for BLinfantis due to a lack of available suitable samples. Such effects could potentially be investigated in studies with a higher temporal resolution. In summary, the impact of the maternal microbiome on infant

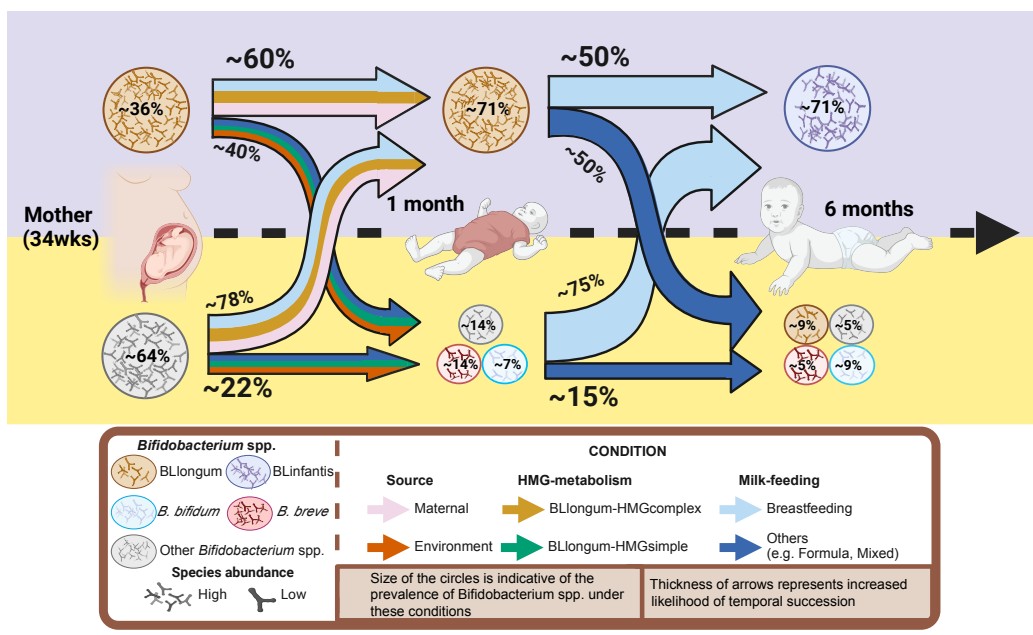

Created in BioRender. Pucci, N. (2024) BioRender.com/f06i661

**Figure 4 Conceptual framework for early-life gut colonization and succession dynamics of *B. longum* subspecies.** For each timepoint (mother, 1 m and 6 m infants) two scenarios are presented: high or low *Bifidobacterium longum* abundance (purple/top and yellow/bottom area). Based on our findings, we highlight the importance of maternal and vertical transmission's impact (pink arrow), breastfeeding (light-blue) and 'complex' HMG-metabolic potential (dark yellow) on BLlongum prevalence and abundance at one month of age. We further postulate that breastfeeding (together with complex and versatile HMG-metabolic potential of BLinfantis) also explain the subsequent takeover of BLinfantis (purple circle) over BLlongum at six months of age. Created in BioRender.

microbiome development was evident and is in line with previous observations about the importance of priority effects (*Sprockett, Fukami & Relman, 2018*; *Ojima et al., 2022*).

Additionally, we found feeding type and frequency to be key factors influencing the dynamics of the infant gut microbiome. Curiously, breastfeeding was an important factor determining BLlongum and BLinfantis abundance at one and six months of age, respectively (as was breastfeeding frequency). This could imply that the more intimate contact between mother and child promotes BLlongum at one month of age. Across longer time scales breast milk consumption seems to promote BLinfantis due to their better HMO and N-glycan scavenging abilities.

The introduction of solid foods (weaning) is known to impact the infant gut microbiome and often starts before six months of age (*Koenig et al., 2011*). This was also true for most AIMS infants, as all but one started weaning by six months of age (Table S1). We had expected BLlongum to thrive on a mixed diet of milk and solid foods, as it is generally better adapted to adult diet and is able to break down complex plant polysaccharides (*Kujawska et al., 2020*; *Komeno et al., 2022*). As this was not the case, we concluded that the sporadic and limited intake of solid food in combination with milk feeding reported by the participating parents may not provide sufficient substrate for BLlongum to outcompete

BLinfantis. However, earlier studies suggest that BLlongum eventually emerges as the main subspecies in the infant gut as solid foods gradually become a more significant source of nourishment for the infant (*Vatanen et al., 2022*).

Our genomic analyses further highlighted the importance of breast milk as nutrition for the microbiome with specific adaptations influencing colonization success and microbiome dynamics. We found all *Bifidobacterium longum* to encode for metabolic pathways associated with metabolic independence that were previously found to be characteristic for successful adult gastrointestinal tract colonizers (see Table S11 metabolic enrichment) (*Vatanen et al., 2022*; *Watson et al., 2023*). BLinfantis genomes encode for a broad set of enzymes involved in (complex) HMO and N-glycan uptake and degradation (*Garrido et al., 2011*; *Vatanen et al., 2019*; *Sakanaka et al., 2019b*; *Arzamasov & Osterman, 2022*), while BLlongum are generally thought to have a limited HMO/N-glycan degradation potential (*Sakanaka et al., 2019a*; *Kujawska et al., 2020*; *Arzamasov & Osterman, 2022*; *Komeno et al., 2022*). Yet, nearly half of BLlongum strains were predicted to potentially uptake and degrade oligomannose-type N-glycans, and the absence of this expanded functional potential was associated with lower BLlongum abundance in one-month infant stool samples, a relationship that had not been reported before.

Though we are unable to draw conclusions about the potential health benefits of different BLlongum HMG-utilization types for infants, our findings indicate the existence of defined succession trajectories involving the different *B. longum* subspecies which are influenced by multiple factors (Fig. 4). Previous studies observed fine deviations in gut microbiome progression during developmental stages to be associated with islet autoimmunity or type 1 diabetes (*Stewart et al., 2018*; *Vatanen et al., 2018*). Future studies should therefore investigate the fate of gut microbiome assembly and address potential health consequences associated with different *B. longum* colonization trajectories.

Leveraging rich datasets in combination with metagenomics and ML allowed us link the unveiled intra-subspecies genomic variation in *B. longum* HMG-utilization potential to the observed early life subspecies replacement. In fact, ML *via* LASSO regressions revealed that the combinations of all three factors (maternal/priority effects, nutrition and bifidobacterial metabolic potential) governed the colonization and microbiome dynamics we observed and allowed us to conceive a mechanistic model (Fig. 4). Though the importance of these factors differed, not one individual factor would have been able to explain the observed dynamics by itself. Nevertheless, we acknowledge the small sample size of the available data does not allow for generalized predictive modeling, yet all associations we detected are congruent with the biological and ecological knowledge.

## CONCLUSIONS

In conclusion, our results highlight the predictable and reversible nature of *B. longum* succession outcomes. Bifidobacteria are considered as the healthy-standard for infant development, with their beneficial impact being the result of extensive bifidobacteria-gut microbiome-host cross talk (*Turroni et al., 2016*). As substrate utilization and metabolite production varies among species, subspecies and even same-species strains, distinct early

life colonization and succession dynamics could potentially influence the benefits they provide. Future long-term longitudinal studies following gut microbiome development from birth to at least three years of age will reveal the potential health implications of differential succession outcomes on infant development, with the potential of designing targeted interventions during a decisive period in human development.

## ACKNOWLEDGEMENTS

We thank Anna Heintz-Buschart, Johan Westerhuis, Vanessa Harris, Egija Zaura and Murat Eren for constructive discussions on the manuscript. In addition, we would like to thank the reviewers for their valuable and insightful comments, which have appreciably contributed to improving the quality of the paper.

### Funding

This project was supported by the GGD Amsterdam, the University of Amsterdam - Research Priority Area Personal Microbiome Health (RPA-PMH), the Stichting Orale Biologie and the Dutch Research Council (NWO) - MetaHealth project (NWA.1389.20.080). The funders had no role in study design, data collection and analysis, decision to publish, or preparation of the manuscript.

### Grant Disclosures

The following grant information was disclosed by the authors:
The GGD Amsterdam, the University of Amsterdam - Research Priority Area Personal Microbiome Health (RPA-PMH), the Stichting Orale Biologie and the Dutch Research Council (NWO) - MetaHealth project: NWA.1389.20.080.

### Competing Interests

The authors declare there are no competing interests.

### Author Contributions

- Nicholas Pucci conceived and designed the experiments, performed the experiments, analyzed the data, prepared figures and/or tables, authored or reviewed drafts of the article, and approved the final draft.
- Joanne Ujčič-Voortman conceived and designed the experiments, authored or reviewed drafts of the article, resources, Project administration, and approved the final draft.
- Arnoud P. Verhoeff conceived and designed the experiments, authored or reviewed drafts of the article, resources, Project administration, Funding acquisition, and approved the final draft.
- Daniel R. Mende conceived and designed the experiments, performed the experiments, analyzed the data, prepared figures and/or tables, authored or reviewed drafts of the article, funding acquisition, and approved the final draft.

## Human Ethics

The following information was supplied relating to ethical approvals (*i.e.,* approving body and any reference numbers):

Methods for sampling and questionnaires received approval by the Medical Ethical Examination Commission of the Amsterdam University Medical Center (METC Amsterdam UMC, Reference number: NL64399.018.17). All participants have provided written consent. Written consent for infants was provided by parents after birth.

## Field Study Permissions

The following information was supplied relating to field study approvals (*i.e.,* approving body and any reference numbers):

Methods for sampling and questionnaires received approval by the Medical Ethical Examination Commission of the Amsterdam University Medical Center (METC Amsterdam UMC, Reference number: NL64399.018.17).

## Data Availability

The dataset supporting the conclusions of this article is available both at the European Nucleotide Archive (ENA) PRJEB66728, and NCBI: 1050518, PRJEB66728.

## Supplemental Information

Supplemental information for this article can be found online at http://dx.doi.org/10.7717/peerj.18602#supplemental-information.

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
