# Peer review of "Priority effects, nutrition and milk glycan-metabolic potential drive Bifidobacterium longum subspecies dynamics in the infant gut microbiome"

_PeerJ, doi:10.7717/peerj.18602_

## Round 0.1 · original submission · Major Revisions

While all reviewers agree that the text is well-written and the experiments are generally well-designed, they have also identified significant areas for improvement that should be addressed. In your response, I would like you to particularly emphasize the following:

1. In-depth analysis of HMO transporters as suggested by Reviewer 2.
2. Clarification of the distinction between HMO utilization and GH groups, also suggested by Reviewer 2.
3. Quantification of priority effects, as requested by Reviewer 1.

Additionally, please pay close attention to the following points:

1. Improvements to the literature overview as noted by Reviewer 1.
2. Addressing inconsistencies with published research as highlighted by Reviewer 2.
3. Correction of typographical errors mentioned by both reviewers.

From my personal review, I found the discussion on weaning versus mixed-fed infants (Lines 497-505) to be unclear. Could you please clarify the definition of 'mixed-fed' in this context? Does it refer to formula intake alongside breast milk, or the intake of solid foods? Were you able to differentiate between these two types of feeding, or other types? Please ensure that this distinction is clear in the main text.

I believe these comments can be addressed effectively, and I am confident that with these revisions, the manuscript will be ready for publication in a short period of time.

Reviewer 1 ·

Basic reporting

The English used in this manuscript is clear and professional for the most part. I have included suggestions to improve how some sentences are worded.
The Introduction is sufficient to explain the context, however, some points in the introduction need additional or different references. If one of the key points of this paper is to examine the influence of priority effects in the infant gut microbiome, I think the authors can add more regarding the ecological background and discussion of priority effects.
The article structure is professional. Figures, tables, and raw data are shared. The resolution of the main figures could be improved. Results to address the question/hypothesis of what factors influence Bifidobacterium longum abundance in the infant gut is shown in this study.

Experimental design

This study seems to be within the aims and scopes of this journal. The methods are sufficient for replication. The authors seek to determine the factors that enable Bifidobacterium longum to persist in the gut, and they find that a combination of several factors such as HMO utilization ability, priority effects, and feeding mode of infant contributes to B. longum persistence. Their conclusions regarding the role of priority effects need a little more analysis.

Validity of the findings

The authors are aware that their sample size from the AIMS cohort is on the smaller side, but they add data from publicly available datasets to add robustness to the study. I did have one comment about the sample size, which I have included in the comments below.
It might help if the authors added the abundance data of each species from their gut microbiome data (for example, the data that Figure S4 is based on with the abundance of other species) for readers to analyze/replicate their findings.
I have also added some comments on additional analyses the authors may want to consider.

Additional comments

General Comments:
In this study, the authors examine the factors such as that could determine bifidobacterial abundance within the infant gut microbiota and conclude that a combination of factors (formula vs breastfeeding, the human milk oligosaccharide (HMO) utilization ability of each bifidobacterial strain, as well as the timing of arrival of each species into the gut ecosystem) are at play. They examine the HMO-utilization abilities of bifidobacteria at the strain level to see how different phenotypes are represented at different time points. I think the strength of this study is that they collected and analyzed samples from mother-infant pairs. The addition of in vivo gut microbiome data that look at the same individual over more than one time point is a useful contribution to the literature. The additional analysis of several publicly available datasets adds robustness to this study. Also, Figure 4 summarizes the findings of this study well. However, after reading this paper I am not convinced that the authors have enough background and analysis to discuss whether priority effects are operating. Therefore, I recommend a major revision before this paper is considered for publication. I have pointed out issues with the citations below, but I would also like to encourage the authors to double-check their citations, both in terms of formatting and relevance. The resolution of the figures could also be improved.

Other comments:

Unless the authors are intentionally using both “microbiome“ and “microbiota” within the text, it might be better to either use one or the other term. If intentional, the differentiation is not clear to me.

L68: The authors should read and cite the review by Fukami T. 2015 in Annual Review of Ecology, Evolution, and Systematics.

L69-70: Citations 14 and 15 do look at bifidobacterial metabolism, but they do not experimentally examine the competitive ability of the strains as this sentence claims. Ojima et al. 2022 ISME J and Laursen et al. 2023 ISME J might be a better source for this sentence, as they look at both the bifidobacterial metabolism of HMOs and the timing of administration. O’Brien et al. 2021 Pediatric Research is a long-term study that follows a cohort of infants given B. infantis as a probiotic, and it shows how the combination of early intervention and human milk oligosaccharides is beneficial for Bifidobacterium colonization in the gut. Another review Sakanaka et al. et al. 2020 in Nutrients gives a comprehensive overview of the enzymatic machinery that bifidobacteria possess for the utilization of HMOs.

L86: The references are italic

L88, L91, L335, L418, L500, L512 etc: Please check the formatting of the citations. For example, [23]-25].

L95-98: This sentence is a little wordy and hard to follow. Could we rephrase it? For example, I understood the sentence to mean “Hence, considering intra-specific genomic diversity, host life-history traits, and environmental heterogeneity is crucial for understanding the process of early gut colonization by Bifidobacterium longum.”
L100-103: This sentence is also a little hard to follow. In L101 where it says “detailed description,” does that mean the metadata of the dataset? Could we rephrase it to say something like, “In combination with rich datasets that provide detailed descriptions, such tools can uncover key factors and quantify their importance in dynamic systems like bifidobacterial populations in the infant gut.”

Methods: Could the authors elaborate more on how B. longum subsp. longum and B. longum subsp. infantis were differentiated in the analysis? (I find that many studies have difficulties doing this, and Ennis et al. 2024 Nature Communications addresses this). For example, did the authors use subspecies-specific gene markers?
Figure 1: I realize some samples did not yield enough quality DNA for analysis after extraction. However, if the goal of this figure is to talk about the succession of bifidobacterial species over time, it would make it easier to compare and see the change in abundance of each species if the data was aligned and ordered by individual. I see hierarchical clustering mentioned for grouping different B. longum subgroups in the methods and for Figure 2, but, it is not clear how or why the data for Figure 1 was clustered here. If the authors wanted to show how different groups of individuals had different dominant taxa, a different visualization method like PCA or NMDS might work here.

L398, L401: I believe the proper format for alpha-L-fucosidase is to use a small capital for the L.

L437: What does “mnt” indicate? Is it an abbreviation for “month”? From the context of the sentence, maybe it should be 1 month?

Using the LASSO regressions is a good way to identify factors that could indicate that priority effects are operating, but I think this section could use more analysis. Could the authors quantify the strength of priority effects? For example, Figure 3 of Debray et al. 2022 in Nature Reviews Microbiology provides possible approaches for detecting priority effects in microbiome data. Vannette and Fukami 2014 in Ecology Letters also provide a way to quantify priority effects.

L443: This sentence might need to be reworded, as it sounds like higher B. longum abundance is what causes priority effects.

L429: Is there a reason why the six-month models did not use publicly available data? Is it because the other datasets did not collect data at the same time point?

L480: Should there be an apostrophe there?

L484: Could the authors elaborate on this sentence? From Figure 3, it looks like B. longum abundance in the mother is the factor with the highest importance. Did the authors mean to say that the relative contribution of maternal strain sharing becomes of lower importance as the infant ages? Or that maternal strain sharing is of lower importance than the presence of other species by 6 months?

L487: Could the authors elaborate on this statistic? I don’t think I saw anything in the results section about looking at such correlations. Would a positive correlation between maternal abundance and vertical transmission mean that for a given taxon, a higher abundance in the maternal gut would mean higher rates of vertical transmission?

L504: Why is “subspecies” in italics?

Figure 4: What is the difference between the purple area (top of the figure) and the yellow area (bottom of the figure)?

Reviewer 2 ·

Basic reporting

Overall, the manuscript has a standard structure and is written in clear English. I have one major concern about data availability and some minor comments about references and typos.

Major concern
The manuscript lacks information about data deposition for the Amsterdam Infant Microbiome Study (AIMS). Specifically, there are no links to depositories with raw reads or assembled metagenome-assembled genomes (MAGs).

Minor and typos
• 33: the noun ‘bifidobacteria” is plural; thus, it is not suitable in this context
• 68-70: references 14-16 do not contain any information about priority effects
• 74-75: it is not clear what the authors mean by "simpler HMO analogs"
• 81: bifidobacteria --> bifidobacterial
• 82-84: reference 19 does not contain data that can support this sentence.
• 88: [23]-25]
• 90-91: [14-16], 24-27]
• 91: references 24-27 do not provide good support for “whereas BLLongum displays greater specialization in the degradation of plant-derived carbohydrates” since they do not contain any information about the utilization of respective glycans or mechanistic details about the structure of underlying metabolic pathways. Consider citing PMID: 33613480 instead.
• 93: bifidobacteria --> bifidobacterial
• 101: a detailed description of what?
• 143: -4ºC or 4ºC freezer?
• 161, 166: 29 and 30 are identical citations
• 158: 2x150bl --> 2x150 bp
• 181: (i) only reference 35 corresponds to CheckM, (ii) what versions of CheckM and GUNC were used?
• 187: i.e --> i.e.,
• 222: Bifidobacteria --> Bifidobacterium or bifidobacteria
• 263: the FMT (fecal microbiota transplantation) abbreviation is not explained
• 265: annotated to the CAZy database; "to" seems like an incorrect preposition here
• 266-268: the phrase “amino-acid sequences for 14 genes” is technically incorrect, since genes are regions of DNA, which is composed of nucleotides.
• 290, 317: Bifidobacteria --> Bifidobacterium or bifidobacteria
• 335: [26,55],59,60]
• 355: bifidobacteria --> Bifidobacterium or bifidobacterial
• 361: strains should not be in italics
• Figure 2C: The GH112 enzyme (number 13) is called lacto-N-biose phosphorylase not lacto-N-biosidase phosphorylase
• Figure 2D: legend includes the complete legend of Fig. 3
• 412: MAGs cannot utilize carbohydrates, corresponding strains do
• 418: [56],62].
• 461-462: it is not clear what bifidobacteria diversity means
• 500: [63,64]-65]
•. 508-509: the phrasing "metabolic pathways previously identified as characteristic for successful adult gastrointestinal tract colonizers" is too ambiguous. What specific pathways do the authors refer to?
• 512: [63,64],67,68]

Experimental design

The manuscript describes a comparative analysis of 57 shotgun metagenomic samples from mothers and infants from the AIMS cohort. The authors accessed the taxonomic composition of the samples and recovered metagenome-assembled genomes (MAGs) belonging to the Bifidobacterium longum species. They also performed functional annotation of MAGs with a particular focus on glycoside hydrolases (GHs) involved in the utilization of human milk oligosaccharides (HMOs) and N-linked glycans. Finally, the authors used statistical modeling and machine learning to identify features associated with B. longum abundance dynamics.

Major concerns
When looking at the host glycan utilization potential encoded in Bifidobacterium MAGs, the authors (i) focus only on genes encoding GHs, ignoring genes encoding characterized HMO transporters, and (ii) conflate HMO utilization and N-glycan utilization, two completely distinct groups of metabolic pathways, throughout the text. These issues led to some incorrect conclusions (see the next session for details).

Minor
239-240: What was the rationale for using Euclidian distance for binary data (gene/presence-absence)? The more appropriate metrics for such data are usually Jaccard or Hamming distances
288-290: it is not clear from where the HMO-utilizing GH-encoding gene PCoA cluster assignments come from; this part is not explained in the previous section

Validity of the findings

1) The taxonomic profiling data in the manuscript are in agreement with recent findings (PMIDs: 37816852 and 38291346) that B. longum subsp. longum (BLlongum) can be succeeded by B. longum subsp. infantis (BLinfantis) at ~6 months in infants receiving breastmilk. These observations are very important and emphasize the notion that specific components of breast milk, human milk oligosaccharides (HMOs), pose a strong selective pressure towards the gut microbial community, giving a competitive advantage to well-equipped HMO utilizers like BLinfantis over weak HMO utilizers like BLlongum.

The mechanisms that drive the variability in HMO utilization capabilities within B. longum species have been extensively studied previously. BLinfantis can metabolize various neutral, neutral fucosylated, and sialylated HMO species intracellularly by uptaking them via a set of ABC transporters encoded in H1, FL1/FL2, and lnb (also known as H5) gene clusters (PMIDs: 21423604, 31489370). Most BLlongum strains can utilize only lacto-N-tetraose (LNT) either intracellularly or degrading it outside the cell via extracellular lacto-N-biosidase LnbX (GH136) to lacto-N-biose (LNB) and lactose, which are then transported into the cell (PMIDs: 27756904, 28392148). Some rare BLlongum strains like SC596 or MC10007 have an enhanced capability to metabolize fucosylated HMOs (2'FL, 3DL, LDFT, LNFP I) due to the presence of a gene cluster encoding a transporter for fucosylated HMOs, alpha-fucosidases, and fucose catabolism enzymes (PMIDs: 27756904, 35768643).

However, the section "Targeted glycoside hydrolase (GH) profile analysis reveals inter- and intra-subspecies variability in HMO-utilization potential" did not fully capture these well-known differences in HMO utilization capabilities between BLinfantis and BLlongum and within BLlongum strains. The main reason for that is that the authors focused only on the distribution of 14 groups of glycoside hydrolases (GHs) and did not look at the representation of complete HMO utilization pathways, which include HMO transporters. The latter are as equally important for HMO utilization as GHs in bifidobacteria. While the authors argue that finding HMO transporters in metagenomic data is hard, I do not fully agree with it. The sequences of characterized HMO transporters from BLinfantis and BLlongum are publicly available (see PMIDs: 21423604, 27756904, 31489370, 31888048, 36866565); thus a procedure like blastp against MAG proteomes can identify them.

Additionally, the selected 14 GH groups cover two distinct metabolic processes: (i) the utilization of HMOs found in human milk and (ii) the utilization N-glycans that decorate glycosylated proteins (found in human milk or other dietary/host proteins). These two processes are conflated throughout the text, and the representation of all 14 GH groups is often incorrectly used as a proxy for HMO utilization capabilities:
379-381: We define MAG glycoside hydrolase (GH) profiles as the presence and copy-number of "GH-encoding genes related to HMO-utilization". Hierarchical clustering assignments of MAGs, based on the set of GH gene profiles were used to describe "differences in HMO-utilization capacity" between and within subspecies and the observed BLlongum/BLinfantis replacement.
407-408: We find that "HMO-utilization potential in BLlongum" is significantly associated with distinct sets of functions (BLlongum-GHsimple: 46 functions, p<.05; BLlongum-GHcomplex: 31 functions, p<.05; Table S6).
411-414: Notably, "reduced HMO-utilization capacity" is associated with the adh gene, encoding for an aldehyde dehydrogenase (NAD+). BLlongum MAGs capable of utilizing complex HMO/HMG were enriched in genes included in the csb and kdp operons, encoding for type 1-G CRISPR variant and a high potassium (K+) transporter, respectively.

The GH38/GH125-family alpha-mannosidases, the GH18/GH85-family endo-beta-N-acetylglucosaminidases, and the GH5_18 beta-mannosidases are involved in N-glycan and not HMO metabolism. Therefore, it is incorrect to draw conclusions about HMO utilization potential based on the representation of these GH families. The separation of BLlongum MAGs and genomes into BLlongum-GHsimple and BLlongum-GHcomplex in Fig. 2 is mainly driven by the absence/presence of GHs constituting the N-glycan utilization pathway (GH5_18, GH18, GH38, GH85, GH125). This grouping does not represent the actual differences in HMO utilization capabilities within BLlongum, which are driven by the absence/presence of lacto-N-biosidase LnbX (GH136), the gene cluster encoding the fucosylated HMO utilization pathway (like in the MCC10007 strain), as well the variant of the GltABC transporter (some variants transport LNT and LNB, some only LNB).
For example, GH136-positive MAGs should be clustered together based on encoded HMO utilization capabilities, but in Fig 2A., they are almost evenly split between the BLlongum-GHsimple and BLlongum-GHcomplex groups.

Therefore certain results and conclusions can be challenged:
437-449: Notably, HMO-utilization potential was also among the selected features; BLlongum haplotypes with the capacity to utilize complex HMOs/HMGs were found to be more abundant relative to BLlongum-GHsimple haplotypes (Fig. 3AB).
The BLlongum-complex/BLlongum-GHsimple grouping does not adequately represent differences in the HMO utilization potential of respective strains.
512-515: Yet, nearly half of BLlongum MAGs were predicted to degrade more complex HMOs and HMGs, and this expanded functional potential was associated with a higher BLlongum abundance in one-month infant stool samples, a relationship that has not been reported before.
This claim is too strong. Most MAGs reported in Fig. 2A do not encode the capability to degrade HMOs beyond LNT (+ partially degrade LNFP I/LNH in the case of GH136-positive strains), which is consistent with previous reports.
Additionally, most of the listed MAGs also do not encode the capability to degrade complex N-glycans because they lack GH18-family endo-beta-N-acetylglucosaminidases (bifidobacterial GH85-family enzymes predominantly target oligomannose-type N-glycans, see PMID: 22745059).

2) Figure 2D contains multiple innacurracies regarding GH specificity:
• Reaction 4: bifidobacterial GH29 enzymes do not efficiently release alpha-1,2-linked fucose resides (PMID: 22138995)
•. Reaction 5: the NanH1 enzyme in BLinfantis does not act on HMOs (PMID: 21288901)
• Reaction 7 is driven by a specialist GH29 enzyme not GH95 (10.5458/jag.jag.JAG-2019_0016)
• Reaction 8 is incorrectly labeled as 9
• Reaction 9: GH125 is a specialist alpha-mannosidase that has distinct specificity compared to GH38 enzymes. It does not release alpha-1,2/3-linked mannose residues (PMID: 36443572)
• Reaction 10: GH18 and GH85 enzymes have distinct specificities, and most GH85 enzymes do not act on hybrid and complex N-glycans (PMID: 22745059)
• Reaction 12: GH136-family lacto-N-biosidase LnbX cleaves the bond between -GlcNAc-b1,3-Gal-, not -GlcNAc-b1,6-Gal- (PMID: 23843461)
• Reaction 13: GH112-family phosphorylase acts only on intracellular lacto-N-biose (LNB) and not LNB-containing HMOs (PMID: 15933016)

3) Some functional annotation results require clarification.
417-420: Additionally, we tested for overall functional and metabolic differences between BLlongum and BLinfantis (Table S7) and found the specific genome content of BLinfantis to reflect known differences between B. longum subspecies [56],62].
This statement is too strong ("differences"). The authors' method showed that riboflavin (B2) biosynthesis was the only significantly different pathway that differentiated BLinfantis and BLlongum. Other "classical" differences between these two subspecies are arabinose, fucose, sialic acid, and urea metabolism, and the methodology did not pick these up (PMID: 19033196).

414-415 These MAGs also exhibited higher copy-numbers of genes related to the vital folate biosynthesis pathway.
Which specific genes were in higher copy numbers? Do the authors refer to a known variation between BLlongum and BLinfantis in terms of para-aminobenzoic acid (PABA) biosynthesis (which is required for folate biosynthesis)? BLinfantis has an operon encoding PABA biosynthesis enzymes (Para-aminobenzoate synthase, aminase component (EC 2.6.1.85) and Aminodeoxychorismate lyase (EC 4.1.3.38)), whereas most BLlongum strains lack it.
The core folate biosynthesis pathway is conserved between BLlongum and BLinfantis, and all genes encoding this pathway are single copy one.

---

## Round 0.2 · Minor Revisions

Thank you for the updates to the manuscript. The revisions have greatly improved the overall quality of the paper, in agreement with the reviewer's responses, which you can find below. Before moving forward, there are a few additional tasks that need to be addressed based on the reviewers’ comments:

Introduce Statistical Tests (Reviewer 1):
Please ensure that the statistical tests requested by Reviewer 1 are introduced and discussed in the relevant sections.

Clarify Confusing Sections (Reviewer 2):
Multiple comments from Reviewer 2 highlighted areas of confusion. These sections need to be revised to reduce ambiguity and ensure clarity.

Improve Splitting of Paralogs (Reviewer 2):
Reviewer 2 has requested improvements in the way paralogs are split and discussed. Please review this feedback and make the necessary adjustments to enhance clarity and accuracy in this aspect.

Enhance Readability (Reviewer 1):
Reviewer 1 also pointed out areas where readability could be improved. Please go through the manuscript and implement edits that will improve the flow and make the text more accessible.

Overall, I believe the paper has improved significantly. After these corrections, it should be in a strong position for publication.

Thank you for your attention to these details. I look forward to receiving the revised manuscript.

Reviewer 1 ·

Basic reporting

The authors have addressed the points I have raised regarding basic reporting.
The references and figures have also been improved.

I have some minor suggestions for improving the readability of some sentences.

L19-21: "Bifidobacterium longum is one of the first colonizers of the newborn gut, playing a crucial role in the healthy development of both the host and its microbiome."

L26: "We analyzed newly sequenced gut microbiome samples of mother-infant pairs…"

L542-543: “Maternal strain sharing and the identification of BLlongum as the most abundant Bifidobacterium (sub-)species were both factors that contributed to priority effects.”

Experimental design

I thank the authors for incorporating the suggestions for analyzing priority effects.

Validity of the findings

I appreciate that the authors have made the statement regarding data availability clearer.

Additional comments

Minor Comments
L292: Microbiota → Microbiome
L543: Bifidobacterium should be in italics

Figure S8: Based on the error bars it looks like there is a significant difference, but were the authors able to assess whether the BLlongum abundances in the “Early arrival of BLlongum: Yes” Group were statistically higher than the “Early arrival of BLlongum: No” Group?

Figure S9: To align with the analysis done for Figure S8, could the authors examine whether there is any statistical significance between Yes/No of Mother-Infant Strain Sharing and Yes/No of BLlongum (mother, %) > other bifidobacteria (mother, %)? (Analysis with Fisher’s Exact Test may work here)

Ecological theory and experiments with aquatic systems suggest that dominance/community evenness affects dispersal dynamics. Total abundance in the mother may not be statistically significant, but perhaps BLlongum being the dominant species is important?

Reviewer 2 ·

Basic reporting

NA

Experimental design

NA

Validity of the findings

The authors have answered most of my comments and improved the manuscript. However, upon re-reading, I found some errors in the transporter/GH gene representation analysis that should be fixed before publication.

Primary
Fig. 2A and S7.
First, it needs to be clarified why two different heatmaps (2A and S7) are used for the representation of genes encoding transporters and GHs. These two figures have identical dimensions in terms of genomes (rows) and measurements (columns) but show different representation pictures. It seems to me that Fig. 2 sometimes underpredicts the number of genes belonging to a specific orthologous group (MnaA), whereas Fig. S7 overpredicts this number (NasH). The authors should keep only one of these figures to avoid confusion.
Additionally, the authors should be more consistent in splitting paralogs into distinct orthologous groups and use genomic context (synteny) when needed. Right now, the decision to separate/no to split specific paralogs seems arbitrary. For example, GH33-family enzymes are divided into 2 groups (NanH1 and NanH2), whereas GH29-family are kept together, even though different GH29 alpha-fucosidases in BLinfantis have different substrate preferences. The enzyme encoded in the H1 cluster (Blon_2336) is an alpha-1,3/4-fucosidase involved in HMO utilization, whereas Blon_0248 and Blon_0426 are alpha-1,6-fucosidases encoded together with AfcT and involved strictly in N-glycan utilization.
The paralogs of HmoA encoded in the H1 cluster should not be collapsed together. These transporters (Blon_2350, 2351, 2352, 2354, encoding substrate-binding components) likely have distinct substrate preferences and have historically been separated as distinct entities (PMID: 20802066).

• GltA in BLinfantis is an LNT/LNB/GNB transporter, not just GNB (PMID: 21423604). I assume GL-SPB2 is the ortholog of GltA that does not have affinity for LNT
• The LNnT transporter described in B. breve is called NahS, not NasH. Fig. 2 likely correctly depicts its representation (some BLinfantis can have a NahS ortholog), but Fig. S7 overpredicts its presence. The homologs of NahS with ~40-50 % seq. identity found in BLinfantis and BLlongul are not its orthologs; they are distinct transporters involved in the transport of galactooligosaccharide and not LNnT
• I don't know if BLinfantis and BLlongum can have 2 functional copies of GH112 enzymes. This has only been described in B. bifidum
• The copy number of GH20_v1 in BLinfantis seems too big (4). Most BLinfantis genomes have 3 different GH20 enzymes max (Blon_0459, Blon_0732, Blon_2355), many do not have Blon_0459 orthologs
• Why BLinfantis and BLlongum lack MnaA but have the other elements of the N-glycan utilization pathway? This does not make sense, especially in BLlongum, where mnaA forms an operon with other N-glycan utilization genes. Fig. S7 depicts the representation correctly

Fig. 2D.
Reaction mediated by lacto-N-biosidase (number 14) is still depicted incorrectly. Lacto-N-biosidase is an endo-acting enzyme that cleaves a lacto-N-tetraose molecule right in the middle (-GlcNAc-b1,3-Gal- bond), releasing lacto-N-biose and lactose.
For accurate information on the reaction mediated by lacto-N-biosidase, I strongly recommend the authors refer to the original publication (PMID: 23843461).
Additionally, I suggest adding to the legend explaining that most enzymes depicted in the scheme are exo-acting and release only terminal monosaccharide residues from non-reducing ends. On the other hand, GH136, GH18, and GH85-family enzymes are endo-acting.

Other minor corrections
395: I suggest rephrasing as reported previously [32,59,60,63,64], to emphasize the novelty of the AIMS cohort (e.g., consistent with results for other cohorts).

410-412: I suggest separating prevalence and abundance results. I.e., for (44.4±21.4%, prevalence=31%), it is not fully clear that 44.4±21.4% is the abundance.

472-474: the alpha-L-fucosidase (GH29, GH95), ß-hexosaminidase (GH20) and sialidases (GH33_v1-2) play an important role in the intracellular degradation of complex N-glycans (Fig. 2D).
Potentially play since the activity of these GHs from bifidobacteria on N-glycans has not been experimentally demonstrated yet.

478-479: potentially involved since the specificities of these bifidobacterial transporters have not been experimentally demonstrated yet.

488-489: citations 62, 65, 65 do not discuss the differences between BLlongum and BLinfantis with regards to these pathways. Consider citing PMID: 19033196 instead.

601-602: Yet, nearly half of BLlongum MAGs were predicted to potentially uptake and degrade oligomannose-type N-glycans (Fig. S7).
MAGs cannot uptake or metabolize, but the corresponding strain(s) can.

From responses:
Differences in fucose metabolism were captured in our targeted GH and transporter analysis (Fig. 2, Table S8)
Fucose catabolism is not mediated by HMO transporters and GHs involved HMO or N-glycan metabolism. Fucose is transported by a permease and is metabolized by a specific pathway, which is not captured in Fig.2 or Table S8 (PMIDs: 33330584, 33460787)

---

## Round 0.3 · accepted · Accept

Congratulations on making the manuscript clear, accessible, and meeting all the rigor standards highlighted by the reviewers! Both the reviewers and I agree that it is ready for publication.

There are just a few minor edits requested. For instance, Reviewer 1 noted some possible misreferences to figures and materials in the main text due to changes in figure numbering. Reviewer 2 suggested a few textual refinements to enhance readability and better align with scientific and publishing conventions.

These adjustments can be made during the final PDF proofing stage.

Reviewer 1 ·

Basic reporting

I have no further comments at this time regarding basic reporting.

Experimental design

I have no further comments at this time regarding the experimental design.

Validity of the findings

I have no further comments at this time regarding the validity of the findings.

Additional comments

Thank you for addressing my previous comments and performing the additional analysis.
I would like for the authors to double-check whether all the figures and tables are correctly referenced within the text.
For example, in L564-L585, Figure S8 is referenced, but I believe it should be Figure S15. As supplementary information has been added and figure order may have changed, please check the other figures and tables as well.

Reviewer 2 ·

Basic reporting

NA

Experimental design

NA

Validity of the findings

The authors have answered my questions, and the paper has improved and reads well.

I list some minor corrections that can be addressed during the proof stage.

110: machine learning --> ML
502: GltFGH has not been shown to be involved in LNT uptake
505: change mnaABC to italics
Figure 2A:
• The presence of HmoABC homologs in BLLongum is not plausible. The genes encoding this transporter are a part of the H1 gene cluster in BLInfantis, and this cluster has not been detected in BLLongum yet. Therefore, these hits are likely false positives either due to a sequence identity threshold issue or caused by permease-competent only
• HmoA4 is also encoded in the H1 gene cluster and is a single copy gene in all B. infantis strains described to date
Figure 2D:
• Reaction 3 is too far from the target glycosidic bond
• The term endo-cellular activity is not conventional. Lacto-N-biosidases (GH136) enzymes in BLLongum are endo-acting but extracellular. Lacto-N-biose phosphorylase (GH112, Reaction 15) is not endo-acting but intracellular. The mechanism of phosphorolysis catalyzed by GH112 family enzymes is (relatively) similar to that used by inverting exo-acting glycoside hydrolases. I suggest keeping the figure layout but removing the endo-cellular activity term.

"Moreover, to clearly show that GH18, GH85, GH112 and GH136 act intracellularly, we have made the following changes"
GH18 are GH85, GH136 in BLlongum/infantis are extracellular enzymes (but endo-acting), GH112 is intracellular (not endo-acting).